# Different Predictor Variables for Women and Men in Ultra-Marathon Running—The Wellington Urban Ultramarathon 2018

**DOI:** 10.3390/ijerph16101844

**Published:** 2019-05-24

**Authors:** Emma O’Loughlin, Pantelis T. Nikolaidis, Thomas Rosemann, Beat Knechtle

**Affiliations:** 1Trinity Centre for Health Sciences, St James’s Hospital, Dublin 8, Ireland; emolough@tcd.ie; 2Exercise Physiology Laboratory, 18450 Nikaia, Greece; pademil@hotmail.com; 3School of Health and Caring Sciences, University of West Attica, 12243 Athens, Greece; 4Institute of Primary Care, University of Zurich, 8091 Zurich, Switzerland; thomas.rosemann@usz.ch; 5Medbase St. Gallen Am Vadianplatz, 9001 St. Gallen, Switzerland

**Keywords:** ultramarathon, running, performance, anthropometry, athlete

## Abstract

Ultra-marathon races are increasing in popularity. Women are now 20% of all finishers, and this number is growing. Predictors of performance have been examined rarely for women in ultra-marathon running. This study aimed to examine the predictors of performance for women and men in the 62 km Wellington Urban Ultramarathon 2018 (WUU2K) and create an equation to predict ultra-marathon race time. For women, volume of running during training per week (km) and personal best time (PBT) in 5 km, 10 km, and half-marathon (min) were all associated with race time. For men, age, body mass index (BMI), years running, running speed during training (min/km), marathon PBT, and 5 km PBT (min) were all associated with race time. For men, ultra-marathon race time might be predicted by the following equation: (r² = 0.44, adjusted r² = 0.35, SE = 78.15, degrees of freedom (df) = 18) ultra-marathon race time (min) = −30.85 ± 0.2352 × marathon PBT + 25.37 × 5 km PBT + 17.20 × running speed of training (min/km). For women, ultra-marathon race time might be predicted by the following equation: (r² = 0.83, adjusted r^2^ = 0.75, SE = 42.53, df = 6) ultra-marathon race time (min) = −148.83 + 3.824 × (half-marathon PBT) + 9.76 × (10 km PBT) − 6.899 × (5 km PBT). This study should help women in their preparation for performance in ultra-marathon and adds to the bulk of knowledge for ultra-marathon preparation available to men.

## 1. Introduction

An ultra-marathon is any running event where the total distance is longer than the traditional 42.195 km (ultrarunning.com). The shortest ultra-marathon is 50 km [1]. The most frequently performed ultra-marathons held as distance-limited events are 50 km, 100 km, 50 miles, and 100 miles ultra-marathons (www.ultra-marathon.org).

Participation in ultra-marathons is increasing worldwide [1]. The proportion of women in ultra-marathons was very low at the beginning of the ultra-running movement but is now increasing [1]. It used to be <10% and is now greater than 20%. Nikolaidis and Knechtle reported that 64,432 men and 24,977 women competed in ultra-marathons in 2016, compared to 2 women and 77 men in 1977 [1]. In 100-mile ultra-marathons held in the United States, the proportion of women increased from almost no participant in the late 1970s to 20% since 2004 [2]. This percentage has remained fairly stable at 10–20% in recent years [2,3,4]. In most ultra-marathons, the proportion of women has increased in recent years, such as in the “Badwater” to 19.1%, in the “Spartathlon” to 12.5% [4], and in the “Swiss Alpine Marathon” held in Switzerland to 16% [3].

To our knowledge, only five studies have examined predictors of performance for women in ultra-marathon [2,5,6,7,8]. They all examined the effect of age on female ultra-marathon performance. Hoffman et al. [2] examined 100 mile (161 km) ultra-marathon running competitions held in North America. A retrospective analysis of the results from 1977 to 2008 revealed that the fastest times were produced by the 40–49 year age group for women. Hoffman [6] examined the effects of anthropometric variables in the 161 km race ‘Western States Endurance Run’. Average running speed and body mass index (BMI) were negatively correlated for women (r^2^ = 0.10, *p* = 0.02). BMI varied considerably even among the top finishers, but lower BMI values were associated with faster running times. When Knechtle et al. [9] analyzed finishers of 100 km ultra-marathons in one-year age groups, they found the age of peak performance at 41 years in women considering all finishers and at 39 years in women considering the top 10 finishers. Knechtle et al. [10] found triceps skinfold thickness and personal best marathon time to be associated with 100 km race time for females. Nikolaidis et al. [1] examined 50 km races and analyzed the fastest finishers in one-year age-group intervals, finding that the age of peak running speed was 40 years in women. Overall, women seemed to reach peak performance in ultra-marathon running around 40 years of age, usually at an older age than men, apart from the results of Knechtle et al. [5], which may be explained by a younger cohort of women in the 100 km race. 

Anthropometric, training and experience variables have all been found to predict performance in ultra-marathons for men. Key predictors of a successful ultra-marathon finish for men are age [10,11] and specific aspects of anthropometry such as low body fat [12], low BMI [6], and low limb circumferences [13]. Other aspects include fast personal best running times, extensive previous race experience [14,15], and high running speed and running volume during training [5,11,16]. All of these studies examined 100 km ultra-marathons and 24 h-timed races, apart from Knechtle et al. [17], who included half-marathon and marathon races. Variables of anthropometry (e.g., body height and body mass), training variables (e.g., running speed of training and volume of training), and experience variables (e.g., personal best times and number of races completed) have only been examined once for women in 100 km races, in 19 athletes [12], leaving room for extensively more research for this distance and all other distances of ultra-marathon such as the 62 km Wellington Urban Ultramarathon (WUU2k). Because of the increasing popularity of ultra-marathons below 100 km, these predictors should be examined for men who race distances less than 100 km. 

Despite the increasing popularity of about-50 km ultra-marathons during the last few years [1], only limited information is available regarding the trends in performance and participation, with most of the studies for predictors for ultra-marathon focusing on races greater than 100 km [5,10,11,14,16,18,19,20]. Age and BMI are the main physical variables that have been investigated for prediction of performance for women [2,6], apart from one study examining 19 females in 100 km ultra-marathons which looked at anthropometric, training, and experience variables [21]. Apart from the study above looking at the age of peak performance in 50 km ultra-marathons [1], to our knowledge, no study has investigated the predictors of performance in the 62 km ultra-marathon and less than 100 km distance races, of which there are many in New Zealand and globally at the moment. 

It was the primary aim of this study to investigate the training and anthropometric variables that influence the race time of women and men in 62 km ultra-marathon and create an equation to predict the race times for females and males. On the basis of the existing literature for women, it was assumed that older age would be a factor for women’s performance. Regarding the available literature for men, it was hypothesized that prior experience, low BMI, and high running speed in training would be associated with the ultra-marathon race time.

## 2. Materials and Methods

### 2.1. Ethical Approval

The study was reviewed and approved by the New Zealand Health and Disability Ethics Committee. Informed consent was gained from participants after verbal and written explanation of the study was given. 

### 2.2. Participants

The organizer of the Wellington Urban Ultramarathon was contacted by e mail and asked if the study could be conducted at the race in 2018. An e-mail was sent out to all athletes before the race, explaining that the author would be at race to check in with consent forms and questionnaires. 

At race check in, the author explained the study at the race briefing, and consent forms and questionnaires were made available to be filled out. All athletes completing the 62 km ultra-marathon were eligible to participate. Anthropometric characteristics, training characteristics, and pre-race experience variables were all determined before the race. Considering anthropometry, training, and experience as predictive variables, the literature in the area, as mentioned above, was examined. In addition, variables including body weight, body height, volume of running training kilometers per week, average running speed of training, and personal best times (PBTs) in different length races were considered to understand whether they would be predictive of ultra-marathon performance. A questionnaire was developed to ask about these characteristics. These independent variables were bi-variately and multi-variately correlated with ultramarathon race times as the dependent variable. 

The WUU2K (pronounced ‘Woo-Too-Kay’) or ’Wellington Urban Ultra 2 K’, is a trail-running endurance event around the hills that surround the capital city of New Zealand. Included in the event is a 43 km marathon, as well as a 62 km Ultra-Marathon. The ‘2 K’ stands for the 2000 m elevation over the 43 km race, which is closer to 3000 m for the 62 km race. The race is run in mid-winter. There is a mix of terrain including single-track trails, mountain-bike trails, and rocky roads. The weather on the day 14 June 2018 was a clear mid-winter day with lowest temperature of 10 °C and highest temperature of 15 °C at midday. Wind speed averaged 26 km/h. Weather data were obtained from www.timeanddate.com/weather/new-zealand/wellington/historic?month=7&year=2018

The organizer provided food and fluids at several aid stations. From a total of 114 starters, 83 took part in the study. This included 57 men and 26 females. At race check in, the participants were informed about the questionnaire and gave their informed written consent. Among the study subjects, all participants finished the race within eleven hours. 

### 2.3. Measurements and Procedures

On the afternoon and evening of the day before the race, a questionnaire and a letter of consent were distributed at race check in in Wellington June 2018 to all participants in the 62 km race. These questionnaires consisted of 51 questions including anthropometric data, training history, and previous race experience. The questionnaire included 51 questions, two questions asking each person to record their actual anthropometric data (e.g., body height, body weight), 23 questions asking for their training data in the pre-race preparation in the past 12 weeks (e.g., average running training volume per week, average running volume of training on trails, average running volume of training on asphalt, average running speed of training, amount of swim and cycle training), and 10 questions examining prior experience (e.g., number of marathons raced previously, lifetime PBTs for marathon, half-marathon, 10 km, and 5 km). Questionnaires were filled in on the spot and given back to the author. These were filled in by race number. The race director then e-mailed the author race time results organized by race number after the event. 

### 2.4. Statistical Analyses 

Data were checked for normality by Shapiro Wilks normality tests, and normally distributed data are presented as mean + SD. All race times (e.g., ultra-marathon race time and reported PBTs) were converted from h–min to min. Race times (min) were converted to average speed (km/h) throughout the race and presented as mean ± SD. The coefficient of variation of performance (CV% = (SD/mean) × 100) for total race time was calculated. Independent t-tests were used to examine the statistical significance between women’s and men’s weekly running volume, longest training run, years running, and longest distance previous race. Pearson correlations were used to investigate potential correlations between the characteristics of anthropometry, training, and previous performance. To reduce the variables for multivariate regression analysis, bivariate analysis was performed between all independent variables, such as age, BMI training, and experience variables, and the dependent variable, race time. In a second step, all significant variables after bivariate analysis entered the multiple linear regression analysis. Stepwise multiple regression (forward) analysis was used to determine the best variables correlated with race performance. Variables were added one by one into multiple regressions by most to least significant *p*-values in the bivariate analysis. A probability value of less than 0.05 was accepted as significant for the multiple regression analysis. Predictive variables were then used to create an equation to predict the ultramarathon race time from the anthropometric and training characteristics. The sample at this point was reduced because of non-completed answers (“NA”s) in our dataset. Five men and six women filled in all data needed to analyze the predicted times. Correlation analysis was used to investigate the reliability of predicted marathon race times with respect to effective marathon race times for these 11 subjects. Bland Altman plots were created to compare differences of means and actual limits of agreement between actual and predicted times of the athletes. All statistics were performed using Open Source Software R and R Studio (RStudio Team (2015). RStudio: Integrated Development for R. RStudio, Inc., Boston, MA, USA).

## 3. Results

### 3.1. Participation 

A total of 26 women and 57 men completed the WUU2k within 9 h 17 min ± 1 h 07 min (CV 12%) and 8 h 43 min ± 1 h 23 min (CV 17%), respectively. The running speed of both women and men were significantly faster than the running speeds during training: 8.8 min/km for women during the race, compared to 6.1 min/km during training; 8.3 min/km for men during the race, compared to 5.8 min/km during training. Table 1 shows anthropometric, training, and experience characteristics of women and their association with race time. Table 2 shows anthropometric, training, and experience characteristics of men and their association with race time. 

Women had a lower body mass, a shorter body height, and a lower BMI compared to men (Table 1). Women had a lower average weekly running volume (65.5 ± 20.0 km/week) compared to men (70.9 ± 22.7 km/week), *p*-value = 0.42. The women’s average longest run (45.5 ± 12.8 km) was similar to men’s (41.5 ± 10.4), (*p*-value = 0.88). Women also had nearly as much experience as men and had been running for a similar amount of time (8.8 ± 5.7 years versus 9.3 ± 8.3 years, *p*-value = 0.76). Women also had previously completed slightly longer distance races than the men (102.0 ± 53.6 km versus 93.9 ± 58.6, *p*-value = 0.55). Women reported slower marathon PBTs than the men (223.2 ± 27.3 min vs. 210.1 ± 25.9 min, *p*-value = 0.097) and reported slower speed of training than men (6.13 min (± 1.1 min)/km, 5.8 min (± 0.9 min)/km, *p*-value = 0.16). 

### 3.2. Bivariate Analysis 

In bivariate analysis, the variables volume of weekly running training (km) and PBT in half-marathon, 10, and 5 km were all associated with race time for women (Table 1). Age, BMI, years running, running speed during training, PBT in marathon and in 5 km were all associated with race times for men (Table 2). 

### 3.3. Multivariate Analysis 

For women, ultra-marathon race time might be predicted by the equation (r^2^ = 0.83, adjusted r^2^ = 0.75, SE = 42.53 on six degrees of freedom) ultra-marathon race time (min) = −148.83 + 3.824 × PBT in half-marathon running + 9.76 × PBT in 10 km running ± 6.899 × PTB in 5 km running. The predicted race time did not correlate significantly (cor 0.82, *p*-value = 0.09) with the achieved race time (Figure 1 (a)). For men, ultra-marathon race time might be partially predicted by the equation (r^2^ = 0.44, adjusted r^2^ = 0.35, SE = 78.15 on 18 degrees of freedom) ultra-marathon race time (min) =  −30.85 ± 0.2352 × PBT in marathon running + 25.37 × PBT in 5 km running + 17.20 × running speed of training. The predicted race time correlated significantly (cor = 0.84; *p*-value = 0.03) with the achieved race time (Figure 1b). Figure 2 shows the level of agreement using the Bland–Altman method (95% limits of agreement −71.0 to 81.1 min) between effective and predicted race time for women. Figure 3 shows the level of agreement using the Bland–Altman method (95% limits of agreement −112.2 to 106.0 min) between effective and predicted race time for men. 

## 4. Discussion

The aim of this study was to investigate whether age or other basic anthropometric characteristics (e.g., body height, body weight, BMI), training characteristics (e.g., weekly volume of training, speed of training) or previous experience (e.g., years running, personal best times in shorter running races) were related to ultra-marathon race time in women and men using bi-variate and multi-variate analyses. On the basis of the existing literature, it was assumed that different predictor variables for women and men would be found.

For women, it was estimated that older age would be a factor for women’s performance. Considering the available literature for men, it was assumed that prior experience, low BMI, and high running speed in training would be associated with the ultra-marathon race time. In contrast to the study hypothesis, it was shown that age was not the most important variable for women competing in ultra-marathon. The hypothesis for males was confirmed, as low BMI, speed of training, and a fast marathon PB were associated with ultramarathon race time.

Recent studies have shown that predictor variables differ for men and women in ultramarathon and other endurance events such as ironman triathletes [2,22,23,24,25]. Best times in half marathon, 10 km, and 5 km races were important for women, along with volume of training. BMI, speed of training, and PBTs in marathon and 5 km races were important for men. This confirmed our hypothesis that the predictor variables would be different for men and women. Our results are similar to those of Knechtle et al. [23] who reported that, for male ironman athletes, anthropometric variables were important, as percent body fat was significantly associated with total race time. In female triathletes, training volume showed a relationship to total race time, in corroboration of our study.

Another interesting finding was that the coefficient of determination of the models was higher in women (r² = 0.83) than in men (r² = 0.44). For women, the predicted race time did not correlate significantly (cor = 0.82, *p*-value = 0.09) with the achieved race time. For men, the predicted race time correlated significantly (cor = 0.84, *p*-value = 0.03) with the achieved race time. The differences in the coefficients of determination in the models might be explained by differences in anthropometric, training, and experience characteristics between women and men.

A first important finding was that the personal best times in 5 km, 10 km, and half-marathon were the best predictors for female ultra-marathon performance. In the multiple regression models, the personal best half-marathon race time was significantly related to the ultra-marathon race time. Overall, it seems previous experience racing and fast personal best times are very important for ultra-marathon performance. This corroborates the results of Knechtle et al. [12] who examined 19 females in a 100 km ultra-marathon and found that the PBT in a marathon showed the highest correlation coefficient. Studies in other endurance sports disciplines such as triathlon showed personal best times in Olympic distance races were predictive in women for performance at Ironman distance [26].

Personal best times in marathon and in 5 km were associated with ultra-marathon race time for men. This adds to the bulk of knowledge available for males indicating that previous marathon personal best times seem to be a strong and independent predictor variable for ultra-endurance running performance in 100 km [15], 350 km multi-day races [17], and 24 h runs [17]. Previous studies have also shown that the personal best time in shorter races was also a predictor for Ironman race time in recreational male athletes [23,27], and PBT, not anthropometry or training volume, was associated with total race time in a triple-iron triathlon [24]. These findings of PBTs and high speed of running during training predicting ultramarathon performance reiterate the importance of intensity in training for men racing ultra-marathons.

A recent study examined females racing in a 100 km distance ultra-marathon [12]. They found no association of race time with years running. We corroborate the results of Knechtle et al. [12], as the variable years running was not associated with race time for women in this 62 km race. Rae et al. [28] examined the interaction of aging and racing on ultra-endurance running performance. Rae et al. [28] found that that overall athletes (18 women, 176 men) took approximately four years to reach peak running speed for a 56 km ultra-endurance race. It seemed that, regardless of the age at which the runners completed their first race, a period of about four years was required for the manifestation of adaptations associated with peak running performance during this ultra-endurance event. In our study, the average years running for females was 8.84, and so they were past this initial four years of improvement.

Years running were associated to race time for males in bivariate analysis. This corroborates the results of Rae et al. [28] who studied mostly men (176 males, 18 females) and examined a similar distance to WUU2K (56 km versus 62 km of WUU2K). These findings contrast those of Knechtle et al. [17], who examined multi-day racing male mountain runners, and of Knechtle et al. [15], who examined 24 h race runners. For both of these studies, years running were not associated with ultra-marathon race time. Also, years running were not associated with marathon time for male marathoners [29]. Years running would seem to be more important for shorter runs, and maybe this would have to do with the intensity of training for performance in shorter runs, relying on less volume but more intense training, which would be easier for a non-novice runner.

Recent studies show that age was an important predictor variable in ultra-marathon running [10]. Women’s age was not significantly associated with race time in this 62 km race. This is in contrast to the results of Knechtle et al. [12], showing that age was bi-variately associated with race time for 100 km female ultra-runners. However, it is noted that five of the top six female athletes in our group were less than 36 years old, therefore, slightly younger than the age reported by Nikolaidis and Knechtle [1] for peak performance for 50 km female ultra-marathoners (40 years). A younger female population in the WUU2K (11% 20–30, 52% 31–40, 22% 4–50, 15% >50 years) with respect to that studied by Nikolaidis and Knechtle [1], who reported most finishers were recorded for women in the age group of 40–44 years, may be the reason for such a difference. It seems that the age of peak performance on long running distances increases with increasing distance and/or race duration [30,31]. However newer studies seem to report a younger age for peak performances for female ultra-marathoners. Cejka et al. [7] showed that the age of the fastest annual female ultra-marathoners worldwide from 1960 to 2012 was 35 years in 100 km races. When the fastest women runners in the “Comrades Marathon” were considered in one-year intervals, the age of the fastest runners was 32.75 years [21]. 

Studies by Knechtle et al. [10] and Rüst et al. [11] showed an association between male athletes’ ages and 100 km race times. In general, in ultra-marathons, the age of the best performance for men is 35 years or older [1,20,21,32,33]. It seems that the age of peak ultra-marathon performance for men usually increases with increasing length of the race distance. In 50 km ultra-marathon running, the age of the best performance is 39–40 years [1], while in 100 km ultra-marathon running, the best race times are achieved at the age of 30–50 years for men. In the WUU2K, men’s age was inversely associated with ultra-marathon race time after bivariate analysis. The present study corroborated previous research for shorter ultra-marathons of 62 km, showing that a younger age of men was associated with faster race time. 

Women seemed to self-select into ultra-marathons much more than men, with women with low body mass entering races [6]. Similar to what reported by Hoffman et al. [6], women competing in the WUU2k were significantly smaller and lighter and had smaller BMIs compared with men. Women’s height, body mass, or BMI were not associated with ultra-marathon race time. This is in contrast to the results of Hoffman et al. [6] who examined the 161 km Western States Endurance Race finding that BMI was inversely associated with finishing times for women and men. However, Hoffman did note that, in this race, there was significant variance of BMIs within the top 10 finishers. It is worth noting our female athletes had a small range of BMI, with 80% of females having a BMI between 18 and 24 kg/m^2^. This suggests that women ultra-marathon runners are similar in anthropometric measurements and that improved performance is associated with other variables rather than body dimensions. 

Studies of men in 100 km [10,11,13], 161 km [6] and triathlon races [23] show men’s BMI is inversely related to endurance race times. As mentioned above, the BMI of men who entered the WUU2K varied. Ninety-five percent had a BMI within the range of 20 to 28 kg/m^2^. Men’s BMI was inversely associated with race time. This corroborates other research reported above and highlights the importance of being lean for even shorter endurance events.

Speed of training has not been examined previously in ultra-marathon distance for women, but for full marathon, high running speed in training was associated with a fast marathon race time in recreational female runners [34] and half-marathon female runners [35]. The running speed of training was not significant for women in the WUU2K. This could be explained by reasoning that females who trained a lot on hills in order to train for a race with this elevation profile would have been running at higher intensity than women who trained on flat ground but at a faster pace, and so this comparison is difficult. 

Average running speed during training was associated with ultra-marathon race time for males at different ultra-marathon distances such as 100 km, 161 km, 350 km, and 24 h races [5,11,17]. Running speed of training was significant for men also in the WUU2K. Similar to what found by Knechtle et al. [17] and Vickers et al. [22] who reported the average speed of training and interval training components for men were associated with race performance, our results suggest that training and, especially, intensity might be more important than anthropometry in male ultra-runners.

A previous study by Knechtle et al. [10] reported a significant association between 100 km ultra-marathon time and duration of run training sessions for women. Volume of running during training has also been shown to be a predictor of performance for women at marathon distance [34] but not at half-marathon distance [36]. The weekly running volume of training was significant for women in the WUU2K. It seems from above that it could be argued that the volume of training but not the intensity (speed) of training is more important for 62 km female ultra-marathoners. 

Previous studies have examined the running volume of training in relation to male ultramarathon race time. Knechtle et al. [17] examined race performance of male mountain ultra-marathoners. They concluded that the running speed of training units, but not the volume of training, was associated with ultra-marathon race time (over 350 km) for male ultra-marathoners. Knechtle et al. [37], in contrast, showed that in master ultra-marathoners, weekly running kilometers were related to running times. Our study concluded that the running volume of training did not predict ultramarathon race performance. This may be explained by differences in populations. Knechtle et al. [37] examined masters’ athletes, whereas only 60% of WUU2K athletes were masters’ athletes. Finally, Rüst et al. [12] examined 100 km ultra-marathoners and marathoners and concluded that the volume of training was important for 100 km ultra-marathoners but not for marathoners. Marathoners rely more on speed during training. In conclusion, it seems from the above that the volume of training is more important for masters athletes and longer distance 100 km runners and seems to become unimportant again in the ultra-long endurance events (350 km). 

This study is the first to examine anthropometric, training, and experience variables and their association with ultra-marathon performance for male and female ultramarathon athletes in the same race. This is important, as most of the previous ultra-marathon performance research has focused on male athletes, and female participation is rising [1]. It is important for women to be considered as a separate group, since, as we can see from above, their performance predictors are different from those of men. 

However, a cross-sectional study is limited regarding the influence and effects of anthropometric and training characteristics on race time in runners, since only an intervention trial can answer this question. Another limitation of this study is the lack of fitness testing of these athletes and of physiological variables analyses, e.g., blood lactate levels, which have been shown to predict performance in endurance sports [38], actual skinfolds measures, height and weight. We focused this study on anthropometry, training, and prior experience. Other aspects, such as nutrition [39] and motivation [40], were not considered. Another limit is that all areas of the questionnaire relied on self-reporting. Self-reporting of times and distances in training is a limitation because it is not possible to establish the reliability and precision of such reporting [41], which may be affected by peoples self-perception of their BMIs [42]. 

Few people fully filled in the entire questionnaire and had several NAs in our database. This may be a reason why the r² was low in some calculations. For future research, the reliability of the training data should be enhanced by quantifying the training data using GPS or mobile phone applications with GPS (78% of WUU2k athletes used an app to log their training). Examining nutrition, motivation, fitness testing, and other groups of athletes, e.g., Paralympic athletes [43], could be incorporated into future research. It would also be pertinent to study the physiological stress responses of women versus those of men after shorter distance ultramarathons, which have only been examined for males previously [44]. 

## 5. Conclusions

Personal best times in half marathon, 10 km, and 5 km races predict ultramarathon race time for women. Personal best time in marathon and 5 km races and speed of training predict ultramarathon race time for men. This study will help women in their specific preparation for performance at ultramarathon and to predict their ultramarathon race time. It also adds to the bulk of knowledge available to men for ultramarathon preparation. Future studies should include GPS training data, physiological variables, nutrition, and motivation to increase the coefficients of determination of the models.

## Figures and Tables

**Figure 1 ijerph-16-01844-f001:**
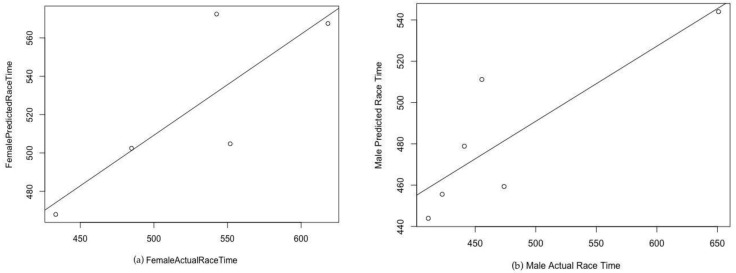
(**a**) Correlation of predicted women’s ultramarathon times with actual race times; (**b**) Correlation of predicted men’s ultramarathon times with actual race times.

**Figure 2 ijerph-16-01844-f002:**
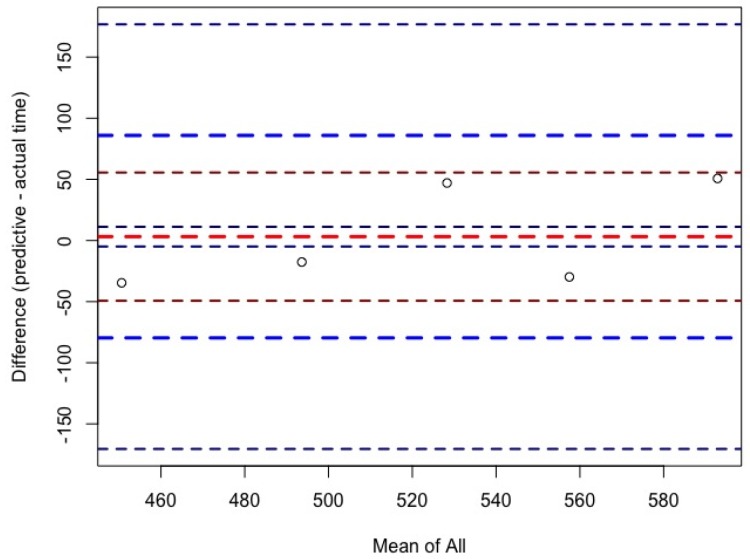
Bland–Altman plots comparing predicted and effective race times for women.

**Figure 3 ijerph-16-01844-f003:**
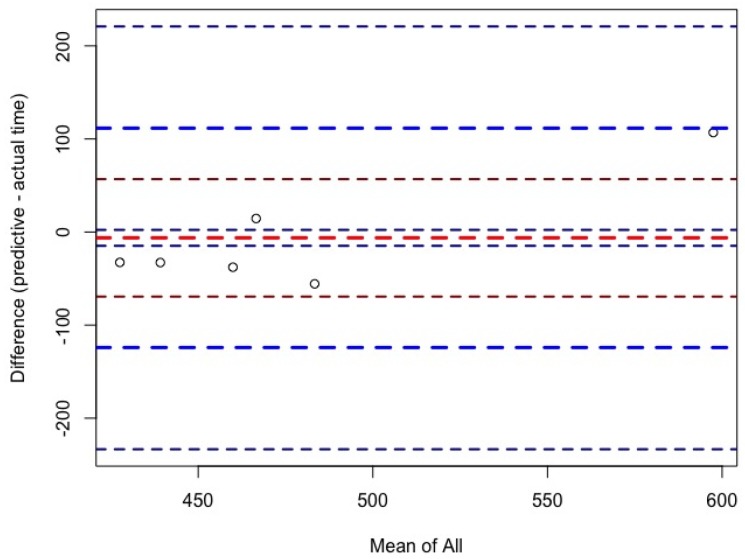
Bland–Altman plots comparing predicted and effective race times for men.

**Table 1 ijerph-16-01844-t001:** Female bivariate analysis (n = 26). Wellington Urban Ultramarathon (WUU), PB (personal best).

	Average ± SD	R Squared	Adjusted R Squared	*p* Value
**Race Finish Time (min)**	557.35 ± 67.23			
**Age**	39.10 ± 1.97	0.049	0.010	0.272
**Body mass (kg)**	61.12 ± 1.16	0.004	−0.037	0.753
**Body height (cm)**	1.66 ± 0.01	0.001	−0.040	0.866
**Body Mass Index (kg/m^2^)**	22.07 ± 0.45	0.004	−0.036	0.732
**Weeks training for WUU2k**	17.88 ±13.26	0.008	−0.033	0.66
**Speed while training (min/km)**	06:08 ± 01:04	0.085	0.031	0.226
**Weekly running distance (km)**	65.52 ± 20.07 *	0.152	0.116	0.049 *
**How many years running**	8.84 ± 5.68	0.011	−0.032	0.614
**Personal best marathon time (min)**	223.16 ± 27.27	0.126	0.074	0.135
**Personal best half-marathon time (min)**	104.91 ± 13.53	0.532	0.509	0.0001 ***
**10 km PB (min)**	45.51 ± 4.73	0.430	0.373	0.021 *
**5 km PB (min)**	22.56 ± 2.86	0.467	0.432	0.002 **

Note: * *p* < 0.05; ** *p* < 0.01; *** *p* < 0.001.

**Table 2 ijerph-16-01844-t002:** Male bivariate analysis (n = 57).

	Ave ± SD	R Squared	Adjusted R Squared	*p* Value
**Race Finish Time (min)**	523.19 ± 89.29			
**Age (years)**	40.59 ± 1.26	0.079	0.061	0.035 *
**Body mass (kg)**	74.35 ± 1.22	0.017	0.0001	0.328
**Body height (cm)**	1.77 ± 0.01	0.042	0.025	0.125
**Body Mass Index**	23.50 ± 0.29	0.102	0.085	0.016 *
**Weeks training for WUU2k**	19.94 ± 15.41	0.0007	−0.01922	0.844
**Speed while training (min/km)**	05:48 ± 00:59	0.200	0.183	0.0001 **
**Weekly running distance (km)**	70.99 ± 22.73	0.003	−0.015	0.701
**How many years running**	9.33 ± 8.29	0.0959	0.079	0.023 *
**Marathon PB (min)**	210.14 ± 266.35	0.255	0.233	0.002 **
**Half-marathon PB (min)**	98.95 ± 26.38	0.064	0.042	0.097
**10 km PB (min)**	44.48 ± 12.64	0.0056	0.029	0.158
**5 km PB (min)**	19.61 ± 2.75	0.249	0.225	0.003 **

Note: * *p* < 0.05; ** *p* < 0.01.

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
