# Peer review of "Different Predictor Variables for Women and Men in Ultra-Marathon Running—The Wellington Urban Ultramarathon 2018"

_ijerph, 2019, doi:10.3390/ijerph16101844_

Round 1
Reviewer 1 Report
This paper presents original objective material, which has value for theory and practice of high performance endurance training. The study design, presentation and consideration raise no objections. One minor remark relates to Bland Altman plots, which need more explanation and comments. Nevertheless the overall estimation of the paper is highly positive.
Author Response
Open Review
(x) I would not like to sign my review report
( ) I would like to sign my review report
English language and style
( ) Extensive editing of English language and style required
( ) Moderate English changes required
(x) English language and style are fine/minor spell check required
( ) I don't feel qualified to judge about the English language and style
Yes | Can be improved | Must be improved | Not applicable | |
Does the introduction provide sufficient background and include all relevant references? | (x) | ( ) | ( ) | ( ) |
Is the research design appropriate? | (x) | ( ) | ( ) | ( ) |
Are the methods adequately described? | (x) | ( ) | ( ) | ( ) |
Are the results clearly presented? | (x) | ( ) | ( ) | ( ) |
Are the conclusions supported by the results? | (x) | ( ) | ( ) | ( ) |
Comments and Suggestions for Authors
This paper presents original objective material, which has value for theory and practice of high performance endurance training. The study design, presentation and consideration raise no objections. One minor remark relates to Bland Altman plots, which need more explanation and comments. Nevertheless the overall estimation of the paper is highly positive.
Answer: We agree with the expert reviewer and have included more explanation and comments regarding Bland Altman plots.
Methods Section: Bland Altman plots were created to compare differences of means and actual limits of agreement between actual and predicted times of athletes
Results Section: Figure 2 shows the level of agreement using Bland - Altman method (95% limits of agreement -71.0 to 81.1 min) between effective and predicted race time for women. Figure 3 shows the level of agreement using Bland- Altman method (95% limits of agreement -112.2 to 106.0 min) between effective and predicted race time for men.
Submission Date
05 May 2019
Date of this review
08 May 2019 17:53:36
Reviewer 2 Report
Introduction
Well designed, information provided according with the final aim of the study
Methods
clearly defined procedure for achieving the proposed objectives
Results
the exposed results respond to the proposed objectives in a rational and clear way
Discussion
discusses the results obtained, although it would be recommended to include the latest studies in the area to improve the quality of the article.
Belinchón-deMiguel, P., Ruisoto-Palomera, P., & Clemente-Suárez, V. J. (2019). Psychophysiological Stress Response of a Paralympic Athlete During an Ultra-Endurance Event. A Case Study. Journal of medical systems, 43(3), 70.
Rubio-Arias, J. Á., Ávila-Gandía, V., López-Román, F. J., Soto-Méndez, F., Alcaraz, P. E., & Ramos-Campo, D. J. (2019). Muscle damage and inflammation biomarkers after two ultra-endurance mountain races of different distances: 54 km vs 111 km. Physiology & behavior, 205, 51-57.
Author Response
Open Review
(x) I would not like to sign my review report
( ) I would like to sign my review report
English language and style
( ) Extensive editing of English language and style required
( ) Moderate English changes required
(x) English language and style are fine/minor spell check required
( ) I don't feel qualified to judge about the English language and style
Yes | Can be improved | Must be improved | Not applicable | |
Does the introduction provide sufficient background and include all relevant references? | (x) | ( ) | ( ) | ( ) |
Is the research design appropriate? | (x) | ( ) | ( ) | ( ) |
Are the methods adequately described? | ( ) | ( ) | ( ) | ( ) |
Are the results clearly presented? | (x) | ( ) | ( ) | ( ) |
Are the conclusions supported by the results? | (x) | ( ) | ( ) | ( ) |
Comments and Suggestions for Authors
Introduction Well designed, information provided according with the final aim of the study Methods clearly defined procedure for achieving the proposed objectives
Results the exposed results respond to the proposed objectives in a rational and clear way Discussion discusses the results obtained, although it would be recommended to include the latest studies in the area to improve the quality of the article.
Belinchón-deMiguel, P., Ruisoto-Palomera, P., & Clemente-Suárez, V. J. (2019). Psychophysiological Stress Response of a Paralympic Athlete During an Ultra-Endurance Event. A Case Study. Journal of medical systems, 43(3), 70.
Rubio-Arias, J. Á., Ávila-Gandía, V., López-Román, F. J., Soto-Méndez, F., Alcaraz, P. E., & Ramos-Campo, D. J. (2019). Muscle damage and inflammation biomarkers after two ultra-endurance mountain races of different distances: 54 km vs 111 km. Physiology & behavior, 205, 51-57.
Answer: We agree with the expert reviewer and have included below studies within the discussion (last paragraph [43] and [44]).
1.Belinchón-deMiguel, P., Ruisoto-Palomera, P., & Clemente-Suárez, V. J. (2019). Psychophysiological Stress Response of a Paralympic Athlete During an Ultra-Endurance Event. A Case Study. Journal of medical systems, 43(3), 70.
2.Rubio-Arias, J. Á., Ávila-Gandía, V., López-Román, F. J., Soto-Méndez, F., Alcaraz, P. E., & Ramos-Campo, D. J. (2019). Muscle damage and inflammation biomarkers after two ultra-endurance mountain races of different distances: 54 km vs 111 km. Physiology & behavior, 205, 51-57.